# The Impact of Age and Vaccine Conspiracy Beliefs on COVID-19 Vaccine Uptake among United States Adults

**DOI:** 10.3390/vaccines12080853

**Published:** 2024-07-30

**Authors:** Victoria A. Furlan, Brian N. Chin, Molly Menounos, Dina Anselmi

**Affiliations:** Department of Psychology, Trinity College, 300 Summit Street, Hartford, CT 06106, USA; victoria.furlan@trincoll.edu (V.A.F.); molly.menounos@trincoll.edu (M.M.); dina.anselmi@trincoll.edu (D.A.)

**Keywords:** health behaviors, health beliefs, social determinants, lifespan, vaccine hesitancy

## Abstract

This observational study examined the relationships between age, vaccine conspiracy beliefs, and COVID-19 vaccine uptake in emerging adults (ages 20–30) and middle-aged adults (ages 50–60) residing in the United States. It also examined sociodemographic predictors of vaccine conspiracy beliefs and COVID-19 vaccine uptake—political conservativism, household income, and educational attainment. We recruited 198 emerging adults and 198 middle-aged adults to complete an online survey assessing vaccine conspiracy beliefs and COVID-19 vaccination status. First, we found that emerging adults reported stronger vaccine conspiracy beliefs than middle-aged adults (estimated mean difference = 0.43, 95CI = 0.08, 0.79, *p* = 0.017), but that emerging adults and middle-aged adults did not differ in their likelihood of being vaccinated with estimated rates of COVID-19 vaccination uptake of 63% in emerging adults and 64% in middle-aged adults. Political conservativism was associated with stronger vaccine conspiracy beliefs and lower COVID-19 vaccine uptake. Lower household income and lower educational attainment were associated with lower COVID-19 vaccine uptake but not associated with vaccine conspiracy beliefs. Second, we found that age moderated the relationship between vaccine conspiracy beliefs and COVID-19 vaccine uptake; stronger vaccine conspiracy beliefs predicted lower COVID-19 vaccine uptake among middle-aged adults (B = −0.63, 95CI = −0.90, −0.36, *p* < 0.001) but were not associated with COVID-19 vaccine uptake among emerging adults (B = −0.21, 95CI = −0.47, 0.05, *p* = 0.12). These results provide insight into the sociodemographic and psychological factors that influence COVID-19 vaccine uptake. Our findings can help to inform the design of targeted public health interventions to increase vaccine uptake in the ongoing fight against COVID-19. Given the crucial role of vaccination in controlling the spread of COVID-19, it is also imperative for future studies to continue investigating how age and vaccine conspiratorial beliefs intersect to impact vaccine uptake.

## 1. Vaccine Conspiracy Beliefs

On 11 December 2020, individuals sixteen years of age and older had access to the first COVID-19 vaccine. While a significant segment of the United States population was hesitant to accept a COVID-19 vaccine before one was available [1,2], Center for Disease Control data show that about 80% of the adult population in the United States had received at least one COVID-19 vaccine dose as of 1 June 2024 [3]. However, nearly 20% of the US population remains unvaccinated against COVID-19. Given that COVID-19 has become endemic, resulting in continuing morbidity, it is important to investigate the factors associated with persistent vaccine hesitancy.

Earlier studies have demonstrated that COVID-19 vaccine hesitancy is associated with demographic and psychological variables including younger age, more conservative political orientation, lower educational attainment, and greater endorsement of vaccine conspiracy beliefs [1,2]. The present study aimed to extend this earlier work by testing the association of vaccine conspiracy beliefs and uptake of the COVID-19 vaccine among emerging adults (ages 20–30) and middle-aged adults (ages 50–60). It also aimed to identify demographic predictors of continued vaccine hesitancy and unwillingness to vaccinate among each age cohort.

Conspiracy theories are often centered around beliefs that powerful groups or organizations orchestrate and control world events, endorsing the idea that events may not always be what they seem or that others cannot be trusted. Although these theories are unsubstantiated and believed by only a small fraction of people [4], recent increases in conspiracy beliefs relating to the medical and scientific domains have led individuals to question the veracity of data and information provided to them by scientists and health professionals, especially relating to vaccine efficacy and safety. Popular vaccine conspiracy beliefs included claims that vaccines are linked to the development of autism or that vaccine efficacy and safety data are fabricated by scientists and pharmaceutical companies [5]. These conspiracy beliefs have, in part, led to increased vaccine hesitancy and anti-vaccine attitudes in the United States. For instance, holding conspiratorial beliefs about vaccines has been found to decrease willingness to vaccinate against COVID-19 [6]. Moreover, exposure to anti-vaccine conspiracies has been shown to reduce a potential parent’s intention to vaccinate their future child [7]. Thus, conspiracy theories may pose a serious threat to public health by promoting mistrust of political and scientific institutions that leads to misinformed and potentially deadly health decisions. In the era of endemic COVID-19, and growing resistance to other vaccines that have historically had higher uptake rates (e.g., childhood vaccines), it is important to understand whether general vaccine conspiracy beliefs predict unwillingness to get the COVID-19 vaccine among different age cohorts.

## 2. Age and Other Sociodemographic Predictors of Vaccination and Vaccine Conspiracy Beliefs

Previous studies have found age differences in vaccination behaviors. Specifically, younger age cohorts have consistently shown lower general vaccination motivation compared to older people, which may be partly explained by their lower perceived danger of diseases [8]. This trend aligns with the current vaccine statistics demonstrating that older Americans have higher COVID-19 vaccination rates than younger Americans. In contrast, research on age and vaccine conspiracy beliefs is limited and inconclusive. Studies exploring the endorsement of both general and COVID-19-related vaccine conspiracy beliefs have found no evidence for age differences [9,10]. However, studies examining the relation between age and endorsement of COVID-19 conspiracies have found a stronger belief in COVID-19 conspiracies among younger individuals [4,11].

Other studies have found associations of political orientation, income, and educational attainment with vaccination behaviors and conspiracy beliefs. Specifically, individuals relying predominantly on conservative media outlets to seek out information on COVID-19 were more likely to endorse COVID-19-related conspiracy theories [12,13]. There is consistent evidence linking a more conservative political orientation with greater belief in both general conspiracy theories [14] and COVID-19 conspiracy theories [15,16]. This tendency subsequently predicted higher COVID-19 vaccine hesitancy among more conservative individuals [14,16].

Studies have also revealed that both lower education and income were associated with lower acceptance of the COVID-19 vaccine prior to its release [17], lower intention and willingness to get the vaccine after its release [18,19], and stronger beliefs in COVID-19 conspiracies [20]. Moreover, lower education levels and vaccine hesitancy may be linked because individuals with lower educational attainment are more likely to display mistrust in scientific research and medical information [21], as well as to endorse COVID-19-related conspiracy theories [11,21].

## 3. Present Study

This cross-sectional study of emerging adults and middle-aged adults in the United States investigated the associations of age, vaccine conspiracy beliefs, and COVID-19 vaccine status. There were two primary aims. The first primary aim was to test whether emerging adults and middle-aged adults differed in their vaccine conspiracy beliefs or their COVID-19 vaccine status. Consistent with earlier research, we hypothesized that emerging adults would be more likely than middle-aged adults to endorse vaccine conspiracy beliefs and less likely than middle-aged adults to be vaccinated against COVID-19. The second primary aim of this study was to examine whether the association between vaccine conspiracy beliefs and COVID-19 vaccine uptake was moderated by age. Because earlier studies suggest that younger individuals are more susceptible to COVID-19 conspiracies than older individuals, we hypothesized that vaccine conspiracy beliefs would be more strongly associated with the uptake of the COVID-19 vaccine among emerging adults than among middle-aged adults.

This study also had the secondary aim of evaluating whether political orientation, educational attainment, and household income were predictive of each age group’s vaccination status and likelihood of endorsing vaccine conspiracy beliefs. Consistent with earlier observations, we hypothesized that individuals in either age group with a more conservative political orientation, lower educational attainment, and lower household income would be more likely to endorse vaccine conspiracy beliefs and less likely to be vaccinated against COVID-19.

## 4. Methods

### Participants and Procedures

We recruited a sample of emerging adults (age range = 20–30 years, mean age = 25.1, SD = 3.1) and middle-aged adults (age range = 50–60 years, mean age = 55.7, SD = 3.2) residing in the United States for an observational study of psychosocial factors associated with COVID-19 vaccine uptake. Participants were recruited through Prime Panels, an online survey platform [22]. Inclusion criteria were age, residing in the United States, and English fluency. There were no exclusionary criteria. Study procedures were approved by the Trinity College institutional review board.

Participants reviewed the consent form and then provided their informed consent by selecting a response option to indicate that they were at least 18 years of age and consented to participate in the study. Next, participants were directed to a Qualtrics survey assessing their vaccine conspiracy beliefs, COVID-19 vaccine uptake, and demographic characteristics. Data were collected from 10 to 31 March 2023; recruitment was halted after data were obtained from 198 participants in each age group.

## 5. Measures

### 5.1. Vaccine Conspiracy Beliefs

A seven-item vaccine conspiracy beliefs scale [23] was used to assess the extent to which individuals held conspiratorial beliefs about vaccines on a seven-point scale from strongly disagree (1) to strongly agree (7). Example items from this scale include “People are deceived about vaccine safety” and “The government is trying to cover up the link between vaccines and autism”. All scale items are provided in the Appendix A. Participants were considered to have valid data for this outcome if they completed at least four of the seven scale items. We calculated a composite variable that represented individuals’ average endorsement of vaccine conspiracy beliefs by averaging the scores of all seven items (α = 0.94).

### 5.2. COVID-19 Vaccination Uptake

Participants were asked to indicate whether they were currently vaccinated against COVID-19; this item was used to assess whether participants were vaccinated or unvaccinated. This was our primary assessment of COVID-19 vaccination uptake that we examined in our main analyses.

Additionally, vaccinated participants were asked whether they were required to get the COVID-19 vaccine and whether they would have gotten vaccinated if not required; these items were used to categorize participants as being voluntarily vaccinated (i.e., vaccinated without requirement; vaccinated by requirement but would have gotten vaccinated anyway) or being unvaccinated or vaccinated by mandate. This was a secondary assessment of COVID-19 vaccination uptake that we examined in our sensitivity analyses.

### 5.3. Covariates

Covariates were selected because of their potential role as third variables in the associations of age cohort, vaccine conspiracy beliefs, and COVID-19 vaccination uptake. These covariates were self-reported gender (male, female, non-binary), race/ethnicity (White, Black/African American, Hispanic/Latino, or other race/ethnicity), educational attainment (less than high school, high school diploma or equivalent, some college/no degree, associate’s degree, bachelor’s degree, more than a bachelor’s degree), household income (continuous: less than USD 35,000 per year, USD 35,000–USD 75,000, USD 75,000–USD 150,000), relationship status (partnered, single, or previously married (divorced/widowed/separated)), presence of a pre-existing health condition (yes, no), and overall political orientation rated from very liberal (1) to very conservative (5).

### 5.4. Data Analysis

We tested Aim 1 by conducting one-way analyses of covariance to examine whether age cohorts (emerging adults/middle-aged adults) differed in their vaccine conspiracy beliefs or their COVID-19 vaccine uptake when controlling for gender, race/ethnicity, educational attainment, household income, relationship status, political orientation, and pre-existing health conditions.

We tested Aim 2 using Model 1 of the PROCESS macro for SPSS to examine whether age cohort moderated the association between vaccine conspiracy beliefs and COVID-19 vaccine uptake when controlling for gender, race/ethnicity, educational attainment, household income, relationship status, political orientation, and pre-existing health conditions. We further investigated statistically significant interactions by examining the covariate-adjusted simple main effect of vaccine conspiracy beliefs on COVID-19 vaccine uptake in each age cohort. We also conducted planned follow-up analyses to evaluate the simple main effect of age cohort on vaccine conspiracy beliefs among vaccinated and unvaccinated individuals using pairwise comparisons of estimated marginal means and standard errors.

Finally, we performed sensitivity analyses to repeat our tests of Aims 1 and 2 using voluntary vaccination as the outcome variable.

## 6. Results

### 6.1. Descriptive Analysis

Our initial sample consisted of 198 emerging adults and 198 middle-aged adults. The demographic characteristics of the initial sample are summarized in Table 1. The average score on the vaccine conspiracy beliefs scale was 3.90 (*SD* = 1.48), and the overall rate of COVID-19 vaccine uptake was 63.9%. There were 18 participants (4.5%) who were excluded from our main analysis because they did not provide complete data for all outcome variables and covariates. This resulted in a final analytic sample of *N* = 378 for all aims.

### 6.2. Aim 1: Testing Age Differences in Vaccine Conspiracy Beliefs and COVID-19 Vaccine Uptake

We tested whether age cohorts (emerging adults/middle-aged adults) differed in their vaccine conspiracy beliefs or COVID-19 vaccine uptake rates when controlling for gender, race/ethnicity, educational attainment, household income, relationship status, political orientation, and pre-existing health conditions.

As shown in Figure 1A, emerging adults reported stronger vaccine conspiracy beliefs than middle-aged adults (estimated mean difference = 0.43, 95CI = 0.08, 0.79, *p* = 0.017). The estimated marginal mean vaccine conspiracy beliefs scale score was 4.12 (*SE* = 0.12) for emerging adults and 3.69 (*SE* = 0.12) for middle-aged adults. In this model, participants who identified as Black (*B* = 0.56, 95CI = 0.16, 0.97, *p* = 0.006) and more politically conservative (*B* = 0.33, 95CI = 0.19, 0.47, *p* < 0.001) also had stronger vaccine conspiracy beliefs (see full model results in Table 2).

As shown in Figure 1B, emerging adults and middle-aged adults did not differ in their COVID-19 vaccine uptake (estimated mean difference = −0.01, 95CI = −0.13, 0.10, *p* = 0.85). The estimated marginal mean rates of COVID-19 vaccination uptake were 0.63 (*SE* = 0.04) in emerging adults and 0.64 (*SE* = 0.04) in middle-aged adults. In this model, participants who were less politically conservative (*B* = −0.09, 95CI = −0.13, −0.04, *p* < 0.001), who had a higher household income (*B* = 0.11, 95CI = 0.05, 0.17, *p* < 0.001), and who had higher educational attainment (*B* = 0.04, 95CI = 0.00, 0.08, *p* = 0.040) had a higher likelihood of COVID-19 vaccination uptake (see full model results in Table 3).

### 6.3. Aim 2: Testing Whether Age Moderates the Association of Vaccine Conspiracy Beliefs and COVID-19 Vaccine Uptake

We tested whether age cohort moderated the association of vaccine conspiracy beliefs and COVID-19 vaccine uptake when controlling for gender, race/ethnicity, educational attainment, household income, relationship status, political orientation, and pre-existing health conditions. Consistent with Hypothesis 2, we found a statistically significant interaction of age cohort x vaccine conspiracy beliefs on vaccination status (*p* = 0.027). Next, we examined the covariate-adjusted simple effect of vaccine conspiracy beliefs on COVID-19 vaccine uptake in emerging adults and middle-aged adults. Stronger vaccine conspiracy beliefs were associated with lower COVID-19 vaccine uptake among middle-aged adults (*B* = −0.63, 95CI = −0.90, −0.36, *p* < 0.001). Vaccine conspiracy beliefs were not associated with COVID-19 vaccine uptake among emerging adults (*B* = −0.21, 95CI = −0.47, 0.05, *p* = 0.12).

Finally, we conducted follow-up analyses to test the significance of age differences in vaccine conspiracy beliefs among vaccinated and unvaccinated individuals. As shown in Figure 2, the estimated marginal mean vaccine conspiracy beliefs scale score was 4.36 (*SE* = 0.18) for unvaccinated emerging adults, 3.97 (*SE* = 0.14) for vaccinated emerging adults, 4.42 (*SE* = 0.18) for unvaccinated middle-aged adults, and 3.29 (*SE* = 0.13) for vaccinated middle-aged adults. Pairwise comparisons indicated that vaccinated emerging adults reported stronger vaccine conspiracy beliefs than vaccinated middle-aged adults (estimated mean difference = 0.58, 95CI = 0.28, 1.08, *p* < 0.001), whereas unvaccinated emerging adults and unvaccinated middle-aged adults did not differ in their vaccine conspiracy beliefs (estimated mean difference = −0.07, 95CI = −0.59, 0.45, *p* = 0.80). Additionally, unvaccinated middle-aged adults reported stronger vaccine conspiracy beliefs than vaccinated middle-aged adults (estimated mean difference = 1.13, 95CI = 0.71, 1.55, *p* < 0.001), whereas unvaccinated and vaccinated emerging adults did not differ in their vaccine conspiracy beliefs (estimated mean difference = 0.39, 95CI = −0.05, 0.82, *p* = 0.09).

## 7. Sensitivity Analyses

We conducted sensitivity analyses that repeated our tests of Aim 1 and Aim 2 with voluntary vaccination as the outcome variable. These analyses compared individuals who were unvaccinated or vaccinated by requirement only to individuals who were voluntarily vaccinated against COVID-19. The results of these analyses were identical to those obtained in the primary analyses.

In sensitivity analyses of Aim 1, age cohorts did not differ in their voluntary uptake of the COVID-19 vaccine (estimated mean difference = −0.10, 95CI = −0.22, 0.02, *p* = 0.10). The estimated marginal mean rates of voluntary COVID-19 vaccination uptake were 0.51 (*SE* = 0.04) in emerging adults and 0.61 (SE = 0.04) in middle-aged adults.

In sensitivity analyses of Aim 2, we found a statistically significant interaction of age cohort x vaccine conspiracy beliefs on voluntary uptake of the COVID-19 vaccination (*p* = 0.032). We observed the same interaction pattern as we did in our main analyses, such that stronger vaccine conspiracy beliefs were associated with lower voluntary COVID-19 vaccine uptake among middle-aged adults (*B* = −0.54, 95CI = −0.78, −0.29, *p* < 0.001), whereas vaccine conspiracy beliefs were not associated with voluntary COVID-19 vaccine uptake among emerging adults (*B* = −0.16, 95CI = −0.40, 0.08, *p* = 0.19). The estimated marginal mean vaccine conspiracy beliefs scale score was 4.25 (*SE* = 0.16) for emerging adults who were unvaccinated or vaccinated by requirement, 4.32 (*SE* = 0.17) for middle-aged adults who were unvaccinated or vaccinated by requirement, 3.97 (*SE* = 0.15) for emerging adults who were voluntarily vaccinated, and 3.30 (SE = 0.14) for middle-aged adults who were voluntarily vaccinated.

Pairwise comparisons indicated that voluntarily vaccinated emerging adults reported stronger vaccine conspiracy beliefs than voluntarily vaccinated middle-aged adults (estimated mean difference = 0.67, 95CI = 0.25, 1.09, *p* = 0.002), whereas emerging adults who were unvaccinated or vaccinated by requirement did not differ from middle-aged adults who were unvaccinated or vaccinated by requirement (estimated mean difference = −0.06, 95CI = −0.55, 0.42, *p* = 0.79). Middle-aged adults who were unvaccinated or vaccinated by requirement reported stronger vaccine conspiracy beliefs than voluntarily vaccinated middle-aged adults (estimated mean difference = 1.02, 95CI = 0.61, 1.44, *p* < 0.001), whereas emerging adults who were unvaccinated or vaccinated by requirement did not differ from voluntarily vaccinated emerging adults in their vaccine conspiracy beliefs (estimated mean difference = 0.29, 95CI = −0.13, 0.70, *p* = 0.18).

## 8. Discussion

Our first aim was to test whether emerging adults and middle-aged adults differed in their COVID-19 vaccine status and in their vaccine conspiracy beliefs. Our hypotheses were partially supported. We found that emerging adults reported stronger vaccine conspiracy beliefs than middle-aged adults, but that emerging adults and middle-aged adults did not differ in their likelihood of being vaccinated or being voluntarily vaccinated. Our second aim was to test whether the association of vaccine conspiracy beliefs and COVID-19 vaccine uptake was moderated by age. Although we found that age moderated the association between vaccine conspiracy beliefs and COVID-19 vaccine uptake, the nature of this moderated effect was the opposite of what we initially predicted. Stronger vaccine conspiracy beliefs predicted lower COVID-19 vaccine uptake among middle-aged adults but were not associated with COVID-19 vaccine uptake among emerging adults. Overall, these findings contribute to prior research by providing novel evidence for age differences in the tendency to endorse vaccine conspiracy theories and in the extent to which vaccine conspiracy beliefs are predictive of COVID-19 vaccine uptake.

### 8.1. Predictors of COVID-19 Vaccination Status and Vaccine Conspiracy Beliefs

Our first hypothesis was partially supported. Emerging adults reported stronger vaccine conspiracy beliefs than middle-aged adults. This observation is consistent with the results of earlier studies showing that younger individuals were more likely to endorse general conspiracies about COVID-19 [4,11], but contrary to the results of other studies reporting no age differences in the endorsement of COVID-19 conspiracy theories [9,10]. We hypothesize that higher social media usage is a likely candidate mechanism that could explain the stronger endorsement of vaccine conspiracy beliefs among emerging adults in this study. If emerging adults were more active on social media, then they may also have had a higher likelihood of coming across misinformation that projects a misguided view of vaccines. Consistent with our proposed explanation, earlier studies have found that younger individuals with higher social media usage, particularly Instagram, Twitter, and YouTube, were more likely to hold coronavirus conspiracies [24]. Future studies could test this hypothesis by evaluating social media usage and exposure to misinformation on social media as mediators of the association between age and vaccine conspiracy beliefs.

Contrary to our hypothesis, emerging adults and middle-aged adults did not differ in their COVID-19 vaccine uptake. This was surprising given nationwide data indicating that vaccination rates are higher for older Americans than for younger Americans [3]. There are several possible reasons why we did not observe evidence for age differences in COVID-19 vaccine uptake in this study. First, the vaccination rate in our sample was lower than the nationwide vaccination rate, possibly because our sample included a greater proportion of individuals who were White, resided in the South, and had lower income and educational attainment. Second, it is possible that our study’s assessment of COVID-19 vaccine status two years after the vaccine was first released was not detailed enough to capture age differences in vaccine hesitancy. For example, it is possible that age differences in vaccine status would have been more evident if we had conducted our study closer to the initial release of the COVID-19 vaccine.

Our second hypothesis was not supported. Although we observed that age moderated the association between vaccine conspiracy beliefs and COVID-19 vaccination status, the direction of this moderated effect was the opposite of what we initially predicted. Specifically, we found that higher vaccine conspiracy beliefs were associated with a lower likelihood of COVID-19 vaccination among middle-aged adults but not emerging adults. Follow-up analyses indicated that this difference was attributable to higher vaccine conspiracy beliefs among vaccinated emerging adults compared to vaccinated middle-aged adults. While vaccinated middle-aged adults had higher vaccine conspiracy beliefs than unvaccinated middle-aged adults, there was no difference in the vaccine conspiracy beliefs of unvaccinated and vaccinated emerging adults. One possible explanation for these findings is that different types of factors may have motivated the decision to receive the COVID-19 vaccination in these age cohorts. Consistent with this proposed explanation, Tanaka et al. [8] reported that, among Japanese people, younger individuals were motivated to receive the COVID-19 vaccination because of concern for others, whereas older individuals were more likely to be motivated to receive the COVID-19 vaccination because of self-interest. It is possible that the motivation to vaccinate against COVID-19 among younger adults was linked to protecting members of the family or the general community rather than protecting oneself [25]. Thus, we speculate that younger adults who believed in conspiracy theories may have been more likely to accept the COVID-19 vaccination because these beliefs were outweighed by prosocial motivations to protect the health and safety of others. This possible explanation is tentative and should be explored in future research. For example, future studies could examine the role of empathy and knowledge about vaccines [26] and altruism and self-interest [27] in promoting or deterring vaccine conspiracy beliefs in different age populations.

### 8.2. Other Predictors of COVID-19 Vaccination Status and Vaccine Conspiracy Beliefs

The secondary aim of this study was to test whether political orientation, income, and educational attainment predicted vaccine conspiracy beliefs or COVID-19 vaccination status. First, we observed that more politically conservative individuals were more likely to endorse vaccine conspiracy beliefs and less likely to be vaccinated against COVID-19. This observation extends earlier evidence that a more conservative political orientation is associated with greater belief in general conspiracy theories [14] and COVID-19 conspiracy theories [15,16] by demonstrating that political conservativism is associated with conspiracy beliefs specifically about vaccines. Second, we observed that income and education were positively associated with COVID-19 vaccine uptake but not associated with vaccine conspiracy beliefs. The first finding is consistent with earlier evidence that lower education and income were associated with lower acceptance of the COVID-19 vaccine before its release [17] and lower intention and willingness to get the vaccine after its release [18,19]. However, the second finding contrasts with an earlier study of Croatian adults which found that both lower education and income were associated with greater belief in COVID-19 conspiracies [20]. However, it should be noted that our study measured general vaccine conspiracy beliefs as opposed to COVID-19 conspiracies. Moreover, Tonković et al. [20] collected their data in 2020, while our data were collected in 2023; it is possible that the varying results may be attributable to our data collection occurring during the endemic era of COVID-19 instead of during the initial phases of the pandemic. More research is needed to explore how educational attainment and income influence endorsement of vaccine conspiracies.

### 8.3. Strengths, Limitations, and Future Directions

The strengths of this study include its examination of two timely and novel research questions and its sensitivity analyses that showed that all associations persist when considering voluntary uptake of the COVID-19 vaccine as the outcome variable. However, there are several limitations to this work that provide opportunities for future research. A primary limitation of this study was the omission of measures assessing participants’ general conspiracy beliefs and their conspiracy beliefs about the COVID-19 pandemic. These measures would have allowed us to examine the relative contributions of general conspiracy beliefs, conspiracy beliefs specific to the COVID-19 pandemic and the COVID-19 vaccine, and general vaccine conspiracy beliefs to COVID-19 vaccine uptake. For example, a study by Jennings et al. [6] reported that belief in general conspiracies, COVID-19 misinformation, and general distrust of vaccines separately contributed to lower acceptance of the COVID-19 vaccine. Follow-up studies could evaluate both general and vaccine-specific conspiracy beliefs and examine their association with COVID-19 vaccine hesitancy and vaccine uptake in younger and older adults. Moreover, there is a general need for follow-up research on the relationship between general conspiracy beliefs and specific conspiracy beliefs regarding the COVID-19 vaccine [4,28]. A second limitation of this study is its use of a self-report questionnaire to assess participants’ vaccine conspiracy beliefs. It is possible that this measure could have been unduly influenced by response biases and other inaccuracies associated with self-report measures. Future studies could address this limitation by testing whether similar patterns are observed when measuring implicit attitudes and beliefs about vaccinations. A third limitation of this study is that we assessed participants’ general conspiracy beliefs about all vaccinations rather than their specific conspiracy beliefs about the COVID-19 vaccination. Future studies could address this limitation by adapting this scale to include items that are specific to the COVID-19 vaccine [29].

Another limitation of this study is its examination of a sample that was primarily White, single, and residing in the Southern region of the United States with lower income and educational attainment. As a result, it is unknown whether these results would generalize to a sample that is more racially, ethnically, socioeconomically, or geographically diverse. For example, Sallam et al. [10] did not observe evidence for age differences in COVID-19 vaccine acceptance and conspiracy beliefs in a study of participants from 14 Arabic-speaking countries. Moreover, Caycho-Rodriguez et al. [29] studied participants from 13 Latin American countries and found that the pattern of age differences in vaccine conspiracy beliefs was different across each country. Future studies are needed to better understand how the intersection of age and cultural factors may influence susceptibility to vaccine conspiracy beliefs.

## 9. Conclusions

The COVID-19 pandemic played a crucial role in fostering conspiracy beliefs about vaccines. This study was among the first to assess the association of COVID-19 conspiracy beliefs with COVID-19 vaccine uptake, rather than intention to vaccinate or attitudes towards the vaccine. Our findings contribute to earlier research by demonstrating that the factors associated with vaccine hesitancy may differ according to age. Future studies are needed to investigate the relationship between vaccine conspiracy beliefs, intention to vaccinate, and vaccine uptake. These findings would inform policymakers and healthcare professionals about how to tailor vaccine promotion interventions to be most effective across different demographic groups for both present and future disease outbreaks.

## Figures and Tables

**Figure 1 vaccines-12-00853-f001:**
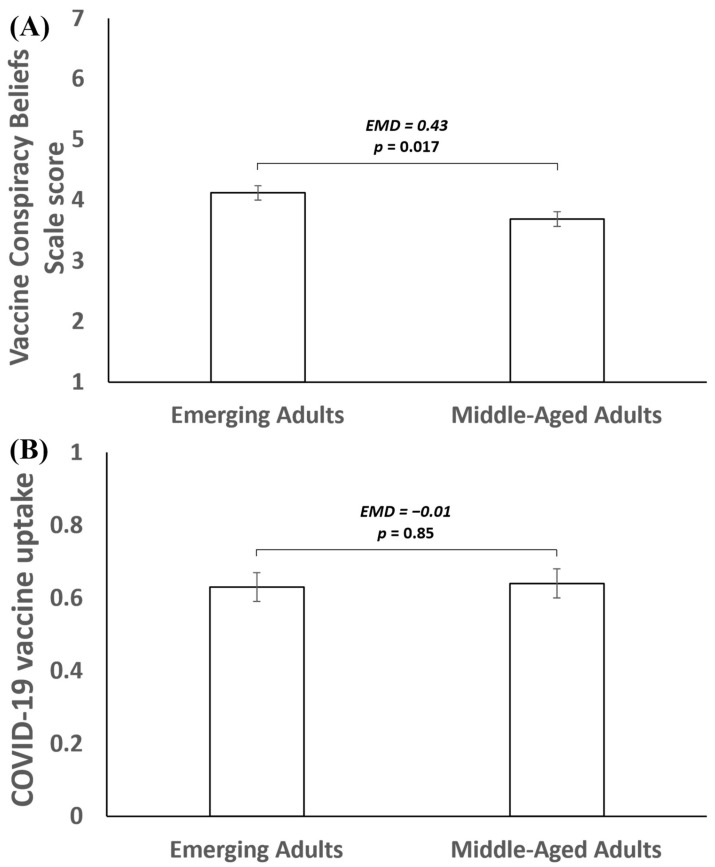
(**A**,**B**) Evaluating the estimated mean difference (EMD) in vaccine conspiracy beliefs (**A**) and COVID-19 vaccination uptake (**B**) by age when controlling for gender, race/ethnicity, relationship status, political orientation, household income, educational attainment, and having a prior medical condition.

**Figure 2 vaccines-12-00853-f002:**
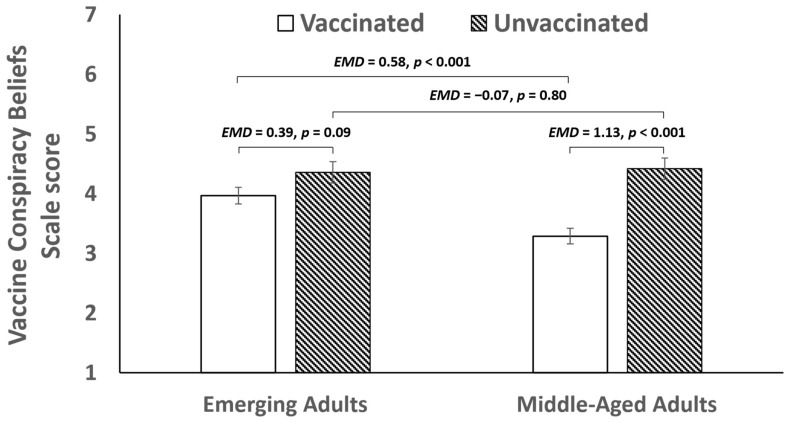
Estimated mean difference (EMD) in vaccine conspiracy beliefs by age and COVID-19 vaccine uptake when controlling for gender, race/ethnicity, relationship status, political orientation, household income, educational attainment, and having a prior medical condition.

**Table 1 vaccines-12-00853-t001:** Demographic characteristics of all study participants (*N* = 396).

Characteristic % (*N*)			
Age			
20–30 years old	50.0 (198)	Overall Political Orientation	
50–60 years old	50.0 (198)	Very Liberal	12.4 (49)
Sex		Liberal	15.2 (60)
Male	46.2 (183)	Moderate	47.0 (186)
Female	53.3 (211)	Conservative	15.4 (61)
Non-Binary	0.5 (2)	Very Conservative	9.6 (38)
Race and Ethnicity		Prefer Not to Disclose	0.5 (2)
White	61.6 (244)	Annual Household Income	
Black/African American	18.9 (75)	<USD 35,000	41.7 (165)
Hispanic/Latino	12.1 (48)	USD 35,000–USD 75,000	35.6 (141)
Asian	2.5 (10)	USD 75,000–USD 150,000	16.2 (64)
Multiple Races	3.0 (12)	>USD 150,000	6.3 (25)
Native American	0.8 (3)	Prefer Not to Disclose	0.2 (1)
Middle Eastern	0.2 (1)	Educational Attainment	
Other	0.2 (1)	Less than high school	3.3 (13)
Prefer Not to Disclose	0.6 (2)	High school or equivalent	34.1 (135)
Region of Residence		Some college (no degree)	29.5 (117)
South	40.9 (162)	Associate’s Degree	13.6 (54)
Midwest	22.7 (90)	Bachelor’s Degree	13.1 (52)
Northeast	20.2 (80)	Graduate Degree	6.3 (25)
West	15.4 (61)	Vaccination Status	
Prefer Not to Disclose	0.8 (3)	Not Vaccinated	35.6 (141)
Marital Status		Vaccinated Voluntarily	56.1 (222)
Single	44.2 (175)	Vaccinated by Requirement	8.1 (32)
Married	24.5 (97)	Prefer Not to Disclose	0.2 (1)
In a Relationship but Not Married	13.6 (54)	Pre-Existing Health Conditions	
Divorced or Separated	13.1 (52)	Yes	23.2 (92)
Widowed	2.8 (11)	No/Unsure	76.8 (304)
Prefer Not to Disclose	1.8 (7)		

**Table 2 vaccines-12-00853-t002:** Parameter estimates for univariate analyses of covariance predicting vaccine conspiracy beliefs.

Parameter	*B*	*SE*	*t*	*p*	95CI Lower	95CI Upper
Intercept	3.43	0.36	9.65	<0.001	2.73	4.12
Female	−0.15	0.15	−0.98	0.33	−0.45	0.15
Non-Binary	−1.75	1.04	−1.68	0.09	−3.80	0.30
Black	0.56	0.20	2.76	0.006	0.16	0.97
Hispanic	0.33	0.24	1.41	0.16	−0.13	0.80
Other Race	−0.20	0.30	−0.66	0.51	−0.80	0.40
Prior Medical Condition	−0.13	0.19	−0.69	0.49	−0.51	0.24
Partnered	−0.22	0.18	−1.23	0.22	−0.56	0.13
Previously Married	−0.24	0.24	−1.02	0.31	−0.71	0.22
Political Conservativism	0.33	0.07	4.58	<0.001	0.19	0.47
Household Income	−0.03	0.09	−0.31	0.76	−0.21	0.15
Educational Attainment	−0.03	0.06	−0.54	0.59	−0.16	0.09
Sample	−0.43	0.18	−2.39	0.017	−0.79	−0.08

Note. Gender, race, and relationship status were dummy coded with reference to participants who were male, White, and single.

**Table 3 vaccines-12-00853-t003:** Parameter estimates for univariate analyses of covariance predicting COVID-19 vaccine uptake.

Parameter	*B*	*SE*	*t*	*p*	95CI Lower	95CI Upper
Intercept	0.51	0.11	4.48	<0.001	0.29	0.73
Female	−0.03	0.05	−0.55	0.58	−0.12	0.07
Non-Binary	0.17	0.33	0.50	0.62	−0.49	0.82
Black	−0.08	0.07	−1.24	0.22	−0.21	0.05
Hispanic	0.03	0.08	0.45	0.66	−0.12	0.18
Other Race	0.12	0.10	1.20	0.23	−0.07	0.31
Prior Medical Condition	0.11	0.06	1.74	0.08	−0.01	0.23
Partnered	0.06	0.06	1.11	0.27	−0.05	0.17
Previously Married	0.08	0.08	1.08	0.28	−0.07	0.23
Political Conservativism	−0.09	0.02	−3.82	<0.001	−0.13	−0.04
Household Income	0.11	0.03	3.66	<0.001	0.05	0.17
Educational Attainment	0.04	0.02	2.06	0.040	0.00	0.08
Sample	0.01	0.06	0.19	0.85	−0.10	0.13

Note. Gender, race, and relationship status were dummy coded with reference to participants who were male, White, and single.

## Data Availability

The data presented in this study are openly available in OSF at https://osf.io/bax5n/ (accessed on 6 June 2024).

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
