# Peer review of "The Impact of Age and Vaccine Conspiracy Beliefs on COVID-19 Vaccine Uptake among United States Adults"

_vaccines, 2024, doi:10.3390/vaccines12080853_

Round 1

Reviewer 1 Report

Comments and Suggestions for Authors

On page 10 you identify "lower income and educational attainment" as a limitation. However, this is not included in your section  8.3. Strengths, Limitations, and Future Directions

These limitations do need to be very clear as it otherwise distorts what are really quite interesting findings.

Author Response

Comment 1: On page 10 you identify "lower income and educational attainment" as a limitation. However, this is not included in your section  8.3. Strengths, Limitations, and Future Directions. These limitations do need to be very clear as it otherwise distorts what are really quite interesting findings.

Response 1: We added a sentence to Section 8.3 to clarify these limitations of our work:

Page 11: “Another limitation of this study is its examination of a sample that was primarily white, single, and residing in the Southern region of the United States with lower income and educational attainment. As a result, it is unknown whether these results would generalize to a sample that was more racially, ethnically, socioeconomically, or geographically diverse.”

Reviewer 2 Report

Comments and Suggestions for Authors

 Thank you for the chance to review this well-written manuscript. Furlan et al. studied vaccine conspiracy beliefs and vaccine uptake among emerging adults (ages 20-30) and middle-aged adults (ages 50-60) in the United States. In addition, the authors examined sociodemographic predictors of vaccine conspiracy beliefs and COVID-19 vaccine uptake. The manuscript is well-written and the statistical analysis is sound. I have a few minor comments that should be addressed before the publication of this article.

The cohort number is not consistent throughout the manuscript. In the Abstract, it says two cohorts, each with 196 participants. In the Methods, it indicates that the initial cohort was 200 participants in each group. Then in the Results, the final total participant number is 198 in each group. Could you please review these numbers and clarify what the initial participant number is?

The authors used a previously validated survey. Reviewing the reference provided, the survey was directed to parents' conspiracy beliefs. How did the authors manipulate the survey to suit their study?

The reference used indicates that the survey was validated for the HPV vaccine. Literature indicates that vaccine conspiracy beliefs differ among different vaccines, with a notable increase in conspiracy theories surrounding the COVID-19 vaccine compared to other vaccines. How can this affect your outcome? Are there more validation studies for this survey that cover other vaccines?

Please add the final version of the survey you implemented as supplementary material.

Discussion: Very well written, but it lacks reference support. Some sentences need to be supported by references. For example: “It is possible that the motivation to vaccinate against COVID-19 among younger adults was linked to protecting members of the family or the general community rather than to protect oneself.” This needs to be supported by a reference. Please add more references to the Discussion where needed.

Limitations: There are several limitations to using a survey for such a study, including response bias and the accuracy of self-reporting. Please add these to the limitations section.

Figures: Please add the EMD abbreviation to the legend.

Could you please add the ethical approval reference number? Also, explain how did you obtain consent from the participants?

Author Response

Comment 1: The cohort number is not consistent throughout the manuscript. In the Abstract, it says two cohorts, each with 196 participants. In the Methods, it indicates that the initial cohort was 200 participants in each group. Then in the Results, the final total participant number is 198 in each group. Could you please review these numbers and clarify what the initial participant number is?

Response 1: We revised the manuscript to include the correct number of 396 participants in the abstract, text, and tables.

Comment 2: The authors used a previously validated survey. Reviewing the reference provided, the survey was directed to parents' conspiracy beliefs. How did the authors manipulate the survey to suit their study? The reference used indicates that the survey was validated for the HPV vaccine. Literature indicates that vaccine conspiracy beliefs differ among different vaccines, with a notable increase in conspiracy theories surrounding the COVID-19 vaccine compared to other vaccines. How can this affect your outcome? Are there more validation studies for this survey that cover other vaccines? Please add the final version of the survey you implemented as supplementary material.

Response 2: Thank you for raising this point. We used the original, unmodified items from the Vaccine Conspiracy Beliefs Scale which assess individuals’ general conspiracy beliefs about vaccinations. We agree with the reviewer that assessing specific vaccine conspiracies beliefs (i.e., relating to the COVID-19 vaccination) could have impacted our findings. We have added several sentences to the discussion to consider this potential limitation:

“A third limitation of this study is that we assessed participants’ general conspiracy beliefs about all vaccinations rather than their specific conspiracy beliefs about the COVID-19 vaccination. Future studies could address this limitation by adapting this scale to include items that are specific to the COVID-19 vaccine.”

We have also added the items from this measure to the supplementary material which we refer to on Page 3 of the main text.

Comment 3: Discussion: Very well written, but it lacks reference support. Some sentences need to be supported by references. For example: “It is possible that the motivation to vaccinate against COVID-19 among younger adults was linked to protecting members of the family or the general community rather than to protect oneself.” This needs to be supported by a reference. Please add more references to the Discussion where needed.

Response 3: We have included several additional supporting references in the revised discussion, including to the sentence flagged by the reviewer.

Comment 4: Limitations: There are several limitations to using a survey for such a study, including response bias and the accuracy of self-reporting. Please add these to the limitations section.

Response 4: We added several sentences to Section 8.3 to clarify these limitations of our work:

“A second key limitation of this study is its use of a self-report questionnaire to assess participants’ vaccine conspiracy beliefs. It is possible that this measure could have been unduly influenced by response biases and other inaccuracies associated with self-report measures. Future studies could address this limitation by testing whether similar patterns are observed when measuring implicit attitudes and beliefs about vaccinations.”

Comment 5: Figures: Please add the EMD abbreviation to the legend.

Response 5: We revised the figure legends to include this abbreviation.

Comment 6: Could you please add the ethical approval reference number? Also, explain how did you obtain consent from the participants?

Response 6: We have added this information to the method section.

Page 3: “Study procedures were approved by the Trinity College institutional review board (Number: 1931).”

“Participants reviewed the consent form and then provided their informed consent by selecting a response option to indicate that they were at least 18 years of age and consented to participate in the study.”

Reviewer 3 Report

Comments and Suggestions for Authors

The topic of the article is uptodate. The new COVID-19 vaccine triggered a debate about the necessity of vaccination and supported to form of new anti-vaccine campaignes and groups and development of conspiracy regarding the vaccines and vaccinations.

The abstract is informative and include all the relevant information. Some exact results can be mentioned.

Introduction: The introduction is divided to sections according to main topics and aims of the article. This structure is logical and the content is appropriate.

Methods: The applied methods was described in detaile. This paert is understandable and contains all relevant information.

Results: In the Table 1 please correct the proportion rate of the 50-60 year old group to 50.0%.

Please correct the decimals that give 100% of the parts in the houshold income and in the vaccination status.

In the Figure 1 and 2 please provide in brackets EMD after estimated marginal means in the note section.

Description of the results are detailed.

Discussion: Discussion is cited all relevant research articles and give explanations for the contradict results.

Author Response

Comment 1: The abstract is informative and include all the relevant information. Some exact results can be mentioned.

Response 1: We have revised the abstract to include some of the study’s exact results:

“First, we found that emerging adults reported stronger vaccine conspiracy beliefs than middle-aged adults (estimated mean difference = 0.43, 95CI = 0.08, 0.79, p = .017), but that emerging adults and middle-aged adults did not differ in their likelihood of being vaccinated with estimated rates of COVID-19 vaccination uptake of 63% in emerging adults and 64% in middle-aged adults.”

“Second, we found that age moderated the relationship between vaccine conspiracy beliefs and COVID-19 vaccine uptake; stronger vaccine conspiracy beliefs predicted lower COVID-19 vaccine uptake among middle-aged adults (B = -0.63, 95CI = -0.90, -0.36, p < .001) but were not associated with COVID-19 vaccine uptake among emerging adults (B = -0.21, 95CI = -0.47, 0.05, p = .12).”

Comment 2: Results: In the Table 1 please correct the proportion rate of the 50-60 year old group to 50.0%. Please correct the decimals that give 100% of the parts in the household income and in the vaccination status.

Response 2: We have made these corrections to Table 1.

Comment 3: In the Figure 1 and 2 please provide in brackets EMD after estimated marginal means in the note section.

Response 3: We revised the figure legends to include this abbreviation.

Reviewer 4 Report

Comments and Suggestions for Authors

Well written study on the relationship between age, vaccine conspiracy beliefs, and COVID-19 vaccine uptake in the USA. A comparison with studies from other countries would be of interest.

Author Response

Comment 1: Well written study on the relationship between age, vaccine conspiracy beliefs, and COVID-19 vaccine uptake in the USA. A comparison with studies from other countries would be of interest.

Response 1: We have added several sentences to the discussion section addressing the need for additional studies testing whether these results would generalize to other countries.

Page 11: “Another limitation of this study is its examination of a sample that was primarily white, single, and residing in the Southern region of the United States with lower income and educational attainment. As a result, it is unknown whether these results would generalize to a sample that was more racially, ethnically, socioeconomically, or geographically diverse. For example, Sallam et al. [10] did not observe evidence for age differences in COVID-19 vaccine acceptance and conspiracy beliefs in a study of participants from 14 Arabic-speaking countries. Moreover, Caycho-Rodriguez et al. [29] studied participants from 13 Latin American countries and found that the pattern of age differences in vaccine conspiracy beliefs was different across each country. Future studies are needed to better understand how the intersection of age and cultural factors may influence susceptibility to vaccine conspiracy beliefs.”